# Effect of Soil Tillage Practice on Photosynthesis, Grain Yield and Quality of Hybrid Winter Wheat

Jan Buczek, Dagmara Migut and Marta Jańczak-Pieniążek *

Department of Crop Production, University of Rzeszów, Zelwerowicza 4 St., 35-601 Rzeszów, Poland; jbuczek@ur.edu.pl (J.B.); dmigut@ur.edu.pl (D.M.)
* Correspondence: mjanczak@ur.edu.pl

**Abstract:** Although the conventional tillage (CT) system dominates in the cultivation of wheat in Europe, currently, mainly for economic and environmental reasons, especially in the case of new varietal genotypes, reduced tillage systems (RT), including no-tillage (NT), are practised. The aim of the research was to evaluate the influence of tillage systems on yield, gas exchange parameters, chlorophyll fluorescence, and the quantity and quality of protein of hybrid winter wheat cultivars grown under various hydrothermal conditions in the years of the research. A field experiment was carried out between 2016 and 2019 in Domaradz (49°47′38″ N, 21°56′54″ E), Poland. The following factors were tested: three tillage systems—conventional (CT), reduced (RT) and no-tillage (NT), and five hybrid cultivars of winter wheat—Hybery, Hyking, Hymalaya, Hypocamp and Hyvento. The highest grain yield and the most favourable values of physiological parameters were found in the CT system in comparison to the reduced systems RT and NT. Unfavourable hydrothermal conditions with a deficit of precipitation during the 2018/2019 growing season resulted in a decrease in the grain yield and selected physiological parameters in the CT system, while they increased in the NT system. More favourable physiological parameters and higher yields resulted from cultivation of hybrid winter wheat in the CT system for cvs. Hymalaya and Hypocamp grain, and in the NT system for cv. Hyking. The use of the CT system in comparison to RT and NT resulted in a significant increase in the value of grain quality parameters and the content of the sum of gliadins and glutenins as well as $\gamma$, $\omega$ gliadins and HMW glutenins. No statistical differences were found in the content of albumin and globulin or $\alpha/\beta$ subunits of gliadins and LMW glutenins in the CT and RT systems. Cvs. Hyvento and Hyking, in the CT as well as in the RT and NT systems, obtained higher values of quality characteristics and fractions and subunits of gluten proteins, especially when low hydrothermal coefficients were recorded during the grain formation and ripening period (June–July).

**Keywords:** hybrid winter wheat; soil tillage; grain yield; gas exchange; chlorophyll content; chlorophyll fluorescence; protein fractions

## 1. Introduction

Wheat is a widely cultivated species in the world with the largest sown area of 215.9 million ha [1]. The high yield and good technological value of the grain make it the most important species sold on international markets. Despite the relatively low protein content of the grain, wheat provides as much protein as the total cultivation of soybeans. This contributes to the high nutritional importance of wheat proteins, especially in less developed countries, where wheat products constitute the basic and significant part of the human diet [2].

The main goal of modern plant breeding is to obtain winter wheat cultivars characterised by yield stability, having the desired quality characteristics and being resistant to adverse environmental conditions, including abiotic stresses caused by climate change [3,4]. Therefore, there is a growing interest in the cultivation and use of hybrid winter wheat grain [5]. Hybrid winter wheat cultivars are distinguished by higher agricultural potential

due to higher yield stability and grain and straw productivity compared to population cultivars, as well as better competitiveness against weeds [5–7]. In addition to increasing the yielding potential, hybrid winter wheat has a more stable productivity under various environmental conditions, including those exposed to stress, compared to traditional pure-line cultivars [8]. Hybrid cultivars are particularly effective under drought stress in regions with low rainfall totals [9]. However, the success of hybrid winter wheat breeding and its implementation is hampered mainly due to the high cost of seed production (due to the low seed set on the male-sterilised female lines) [10].

The main soil tillage system in the world, and especially in Europe, is the conventional plough system [11]. However, due to the large interference in the soil and the cost-consuming nature of the conventional tillage (CT) system, the area of wheat cultivation with the use of reduced tillage (RT) and no-tillage (NT) systems is increasing [12]. Alternative tillage systems (RT, NT) are used mainly in regions threatened by erosion, with repeated semi-drought periods during the growing season and a negative balance of soil organic matter [13].

The reduced tillage system (RT) promotes the improvement of soil structure, increases the organic carbon content in the soil, minimises the risk of soil erosion, conserves soil water and reduces soil temperature fluctuations [14]. The no-tillage (NT) system, by reducing the intensity of cultivation, improves soil quality and reduces labour costs and fuel consumption [15,16].

Reduction in tillage improves the capacity of water storage and use in soil, providing a good soil environment for plant development and photosynthesis. Compared to the conventional system (CT), reductions in tillage lead to a higher level of actual photochemical efficiency of photosystem (PSII) and photosynthetic electron transport capacity, and improve leaf photosynthetic capacity [17]. However, the high proportion of crop residues on the soil surface makes it difficult to apply no-tillage (NT) due to slower drying up and soil warming after cold winters. In addition, it causes a decrease in plant productivity, due to the presence of allelopathic effects of crop residue, a greater incidence of pests and a reduction in the effectiveness of fertilisers and herbicides [18].

The yield and technological value of wheat grain are genetically determined and depend mainly on the properties of the cultivar as well as on agronomic and environmental factors [19]. The quality of the protein is determined by the composition of gluten proteins, including the amount of high (HMW) and low molecular weight (LMW) glutenin proteins as well as polymer $\alpha/\beta$, $\gamma$ and $\omega$ gliadins [20]. According to some authors [20–22], tillage systems do not significantly differentiate the qualitative characteristics and fractional composition of wheat grain protein. The research of other authors shows that conventional tillage (CT), compared to reduced tillage systems (RT, NT), results in a higher content of protein, gluten and sedimentation index as well as gluten protein fractions in the grain, especially when weather conditions are unfavourable for mineralisation of crop residues in RT and NT [23–25].

So far, the literature data on the physiological properties and quality of hybrid winter wheat grain grown in various tillage systems are scarce. Therefore, the aim of this study was to assess the effect of tillage systems on yield, gas exchange parameters, chlorophyll fluorescence and grain quality, including the fractional protein composition of hybrid winter wheat cultivars cultivated under different hydrothermal conditions in the years of the study.

## 2. Materials and Methods

### 2.1. Field Trial Site and Experimental Treatments

A two-factor field experiment was carried out in three seasons, between 2016 and 2019 in Domaradz (49°47′38′′ N, 21°56′54′′ E) in south-eastern Poland. The experiment was carried out in 3 repetitions, in a random block pattern (width 6 m × length 75 m), which were divided into 3 sub-blocks.

The impact of two factors was analysed:

- Factor I—three soil tillage systems (T): (1) conventional (CT); (2) reduced (RT); and (3) no-tillage (NT), (Table 1);
- Factor II—five hybrid winter wheat cultivars (C): Hybery, Hyking, Hymalaya, Hypocamp and Hyvento [26,27], (Table 2).

**Table 1.** Tillage systems—factor I.

| Tillage (T) | Cultivation Measures |
|---|---|
| CT | Shallow ploughing was performed to a depth of 10–12 cm and harrowing, and in the first third of September, after the forecrop had been harvested, pre-sowing ploughing was performed to a depth of 20–22 cm. Shallow ploughing was performed with a 5-furrow and pre-sowing ploughing with a 3-furrow reversible plough. |
| RT | In post-harvest tillage, a disc harrow (2.5 m wide) was used to a depth of 13–15 cm, and before sowing, a tilling set (cultivator + string roller + harrow) 2.5 m wide was used. |
| NT | Herbicide treatment with glyphosate was performed at a dose of 4.0 dm$^3$ ha$^{-1}$ and wheat was sown directly into the stubble with a seeder with double disc coulters. |

CT: conventional tillage; RT: reduced tillage; NT: no-tillage.

**Table 2.** Agronomic characteristics of hybrid winter wheat cultivars—factor II.

| Cultivar (C) | Country Registering [1] | Grain Quality [2] | Maturity [3] | Plant Height [4] | Time of Sowing [5] |
|---|---|---|---|---|---|
| Hybery | FR, PL | B | ML | M/T | VE/O |
| Hyking | BE, CZ, FR | A | E | L/M | E/O |
| Hymalaya | DE | A | M | M | VE/O |
| Hypocamp | BE, FR | B | E | M | E/O |
| Hyvento | DE | A | M | M | O |

[1] BE: Belgium; CZ: Czechia; DE: Germany; FR: France; PL: Poland; [2] A: high quality wheat; B: complementary bread wheat; [3] E: early; ML: medium late; M: medium maturity; [4] L/M: low to medium; M: medium; M/T: medium to tall; [5] VE/O: very early to optimum; E/O: early to optimum; O: optimum.

During all of the study years, winter wheat was sown between 25 September and 5 October at a sowing density of 180 seeds m$^2$ with row spacing 14–15 cm, to a depth of 3–4 cm. Winter oilseed rape was the previous crop in all the years of research.

Mineral fertilisation and plant protection products were applied in the appropriate development stages of wheat according to the BBCH scale (Biologische Bundesanstalt, Bundessortenamt and Chemische Industrie) [28]. Nitrogen fertilisation (NH$_4$NO$_3$) was carried out in the spring after the start of vegetation in a dose of 60 kg ha$^{-1}$, and additionally during the growing season in doses of 60 and 40 kg ha$^{-1}$, in the stages of stem elongation (32–33 BBCH) and heading (54–56 BBCH). Fertilisation with phosphorus (Ca(H$_2$PO$_4$)$_2$) at a dose of 80 kg ha$^{-1}$ and potassium (KCl) at a dose of 120 kg ha$^{-1}$ was applied once in the autumn (CT, RT) or in the spring (NT). Plant protection products were used in accordance with the recommendations of the Institute of Plant Protection—National Research Institute in Poznań, Poland [29].

The herbicides Maraton 375 SC (pendimethalin, isoproturon, BASF SE, Ludwigshafen, Germany) and Aminopielik 450 SL (2,4-dichlorophenoxyacetic acid, ADAMA, Brzeg Dolny, Poland) were used in doses of 4.0 and 3.0 dm$^3$ ha$^{-1}$, and the fungicides Tilt Plus 400 EC (propiconazole, fenpropidin, Syngenta Crop Protection AG, Basel, Switzerland) and Artea 330 EC (propiconazole, cyproconazole, Syngenta Crop Protection AG, Basel, Switzerland) in doses of 1.0 and 0.5 dm$^3$ ha$^{-1}$. Herbicides were used in the tillering phase of wheat (21–22 BBCH) and fungicides in the stem elongation (32–33 BBCH) and heading (54–56 BBCH) phases. Insecticide Bi 58 New EC 400 (dimethoate, BASF SE, Ludwigshafen, Germany) was used at a dose of 0.5 dm$^3$ ha$^{-1}$ in the heading phase (54–56 BBCH), and in the

stem elongation phase (32–33 BBCH), Moddus 250 EC growth retardant (trinexapac-ethyl, Syngenta Crop Protection AG, Basel, Switzerland) at a dose of 0.4 $dm^3\ ha^{-1}$.

### 2.2. Soil and Weather Conditions

The experiment was located in soil originated from silty clay, medium-heavy soil, classified as Haplic Cambisol (CMha) according to [30]. The amount of $N_{min}$ (in 0.01 $CaCl_2$ solution) as measured before wheat sowing varied from 53.8 to 65.0 kg $ha^{-1}$. Soil parameters were determined before establishing the experiment pH 6.15–6.69 in 1 M KCl, and organic matter (with Tiurin's method) 14.0–15.1 g·$kg^{-1}$.

Content of available macroelements in mg $kg^{-1}$ of soil ranged from high to very high for P (66.5–108.1), from low to medium for K (100.1–150.1) and very high for Mg (69.1–75.5). Content of available microelements in mg $kg^{-1}$ of soil were very high for Zn (13.5–15.1) and Cu (5.1–5.8), medium for Mn (82.3–88.5) and low for Fe (550.0–578.1).

The soil samples were determined for contents of available forms of P and K (with the Egner–Riehm method), Mg (with Schachtschabel's method) and micronutrients (with Rinkis's method).

Weather conditions are given according to the University of Rzeszów's Meteorological Station in Rzeszów. Precipitation totals in the study years were more varied than average air temperatures (Figure 1). During the autumn vegetation period (September–November), the weather conditions were favourable; only in the 2018/2019 season was the amount of precipitation lower by 43.3 mm compared to the average for the period. The average temperature during the winter dormancy (December–March) was higher than the average temperature, except for the 2016/2017 season. The spring–summer vegetation period (April–July) in the 2017/2018 and 2018/2019 seasons can be classified as quite dry Sielianinov's hydrothermic coefficients (K = 1.3 and 1.0) and the 2016/2017 season as quite wet (K = 1.9) (Table 3). In the 2018/2019 season, during the period of grain formation and ripening (June–July), the average monthly temperatures were similar (June) or higher by 4.7 °C (July) than the average temperatures, and the rainfall deficit in this period was the highest in the season, amounting to 133.9 mm.

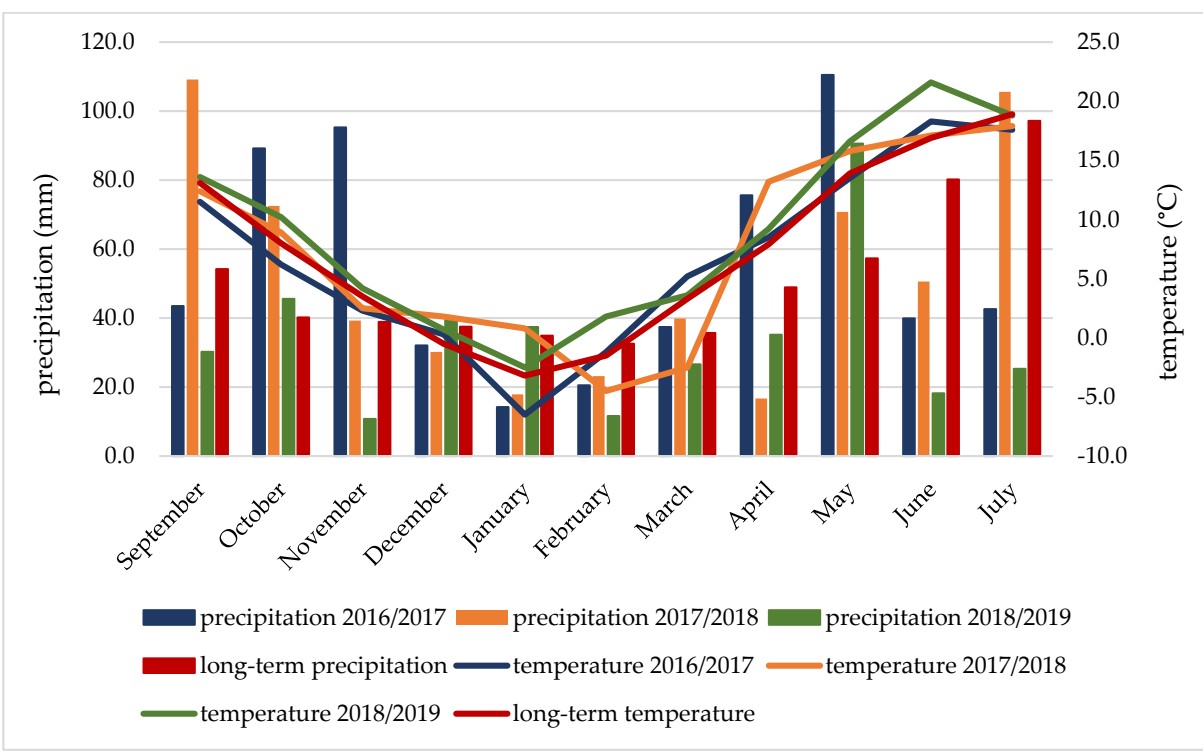

**Figure 1.** Mean temperature and precipitation during the vegetative seasons of 2016/2017–2018/2019.

**Table 3.** Sielianinov's hydrothermic coefficients (K) in vegetation periods of winter wheat.

| Year | Month (K) | | | | K | Sum (IV–VII) | |
|---|---|---|---|---|---|---|---|
| | April | May | June | July | | p | t |
| 2016/2017 | 3.0 (vh) | 2.5 (h) | 1.1 (rd) | 1.0 (d) | 1.9 (rh) | 278.6 | 1478 |
| 2017/2018 | 0.5 (vd) | 1.6 (o) | 0.9 (d) | 2.0 (rh) | 1.3 (rd) | 243.7 | 1928 |
| 2018/2019 | 1.3 (rd) | 1.7 (rh) | 0.4 (ed) | 0.7 (vd) | 1.0 (rd) | 169.0 | 1770 |
| 1956–2015 | 1.7 (rh) | 1.8 (rh) | 1.6 (o) | 1.5 (o) | 1.4 (o) | 283.6 | 1698 |

K—Sielianinov's hydrothermal coefficients (K = (p × 10)/Σt); Coefficient (K) value [31]: ed—extremely dry, vd—very dry, d—dry, rd—rather dry, o—optimal, rh—rather humid, h—humid, vh—very humid, eh—extremely humid; p—precipitation (mm), t—temperature (°C).

*2.3. Physiological Measurements*

Physiological measurements on wheat plants were carried out in the morning in the milk ripe phase (74 BBCH) (13–22 June). Measurements of the content and fluorescence of chlorophyll as well as gas exchange were performed on the flag leaves in the central part of the leaf blade, avoiding the main nerve.

2.3.1. Leaf Area Index

Measurement of leaf area index (LAI) was performed in 4 repetitions using an LAI 2000 instrument (LI-COR, Lincoln, NE, USA) by performing one measurement from above the canopy and 4 measurements inside the canopy.

2.3.2. Relative Chlorophyll Content

Measurements of relative chlorophyll content were carried out on 20 randomly selected plants using a Chlorophyll Content Meter CCM-200plus (Opti-Sciences, Hudson, NH, USA).

2.3.3. Chlorophyll Fluorescence

Chlorophyll fluorescence was measured using a portable chlorophyll fluorescence meter (Pocket PEA, Hansatech Instruments, King's Lynn, Norfolk, UK). Measurements of chlorophyll fluorescence were carried out on 4 randomly selected plants over a 30 min dark adaptation period, using leaf clips [32]. The maximal available intensity was 3500 μmol, which was applied for 1 s with light with a peak wavelength of 627 nm. The parameters analysed in the study were the maximal quantum yield of photosystem II (PSII), photochemistry ($F_v/F_m$), the maximum quantum yield of primary photochemistry ($F_v/F_0$) and the performance index (PI).

2.3.4. Gas Exchange

Net photosynthetic rate ($P_N$, μmol m$^{-2}$ s$^{-1}$), transpiration rate (E, mmol m$^{-2}$ s$^{-1}$), stomatal conductance ($g_s$, mmol $H_2O$ m$^{-2}$ s$^{-1}$) and intracellular $CO_2$ concentration ($C_i$, μmol $CO_2$ m$^{-2}$ s$^{-1}$) were measured simultaneously on a Portable Photosynthesis Measurement System LCpro-SD (ADC BioScientific Ltd., Hoddesdon, UK). In the determination process, the light intensity was 1500 mol m$^{-2}$ s$^{-1}$ and the leaf chamber temperature was 28 °C. Gas exchange measurements were carried out on 4 randomly selected plants. Water use efficiency (WUE) was calculated as $P_N$ divided by E.

*2.4. Grain Yield and Plant Material*

The winter wheat was harvested at full grain ripeness (89–92 BBCH) using a combine harvester. The grain yield from the plots was calculated per 1 ha taking into account 15% moisture content. After harvesting, the wheat grain was dried and cleaned, and then samples were taken for chemical analyses. Grain samples harvested from four replications were blended with a shake shifter and cleaned prior to conditioning and milling in order to obtain enough grain for milling and quality analysis. Wheat grain samples (moisture 13.0 ± 0.1%) were ground in a Quadrumat Junior mill (Brabender, Germany) with a conical screen (mesh size φ = 212 μm) according to AACC Method No. 26-50.01 [33].

### 2.5. Analytical Methods

2.5.1. Quality Testing

Nitrogen content was measured and calculated into crude protein content using the N × 6.25 conversion ratio based on AACC Method No. 46-11.02 [33]. Wet gluten content, gluten index (GI) and falling number were determined in the flour following AACC Method No. 38-12.02 and No. 56-81.03 [33]. Total ash and Zeleny sedimentation were determined in accordance with ICC Standard Method No. 104/1 and No. 116/1 [34].

Crude protein content was evaluated by the Kjeldahl method, wet gluten content by means of a Glutomatic 2200 device (Perten Instruments AB Huddinge, Sweden), sedimentation index by the Zeleny method and falling number by the Hagberg–Perten method on falling number 1800 apparatus (Perten, Huddinge, Sweden).

2.5.2. Protein Extraction and Analysis

Quantitative and qualitative protein characteristics were analysed with the RP-HPLC technique using the solvent system developed by Wieser et al. [35].

Determination of subunits and protein fractions in wheat grain was performed according to Wojtkowiak and Stępień [36].

Albumins plus globulins were twice extracted with 1 mL of 0.4 mol $L^{-1}$ of NaCl with 0.067 of mol $L^{-1}$ $HKNaPO_4$ (pH 7.6); gliadins were extracted with 1 mL of 60% ethanol (three-fold extraction), and glutenins were twice extracted with 1 mL of 50% 1-propanol, 2 mol $L^{-1}$ of urea, 0.05 mol $L^{-1}$ of Tris-HCl (pH 7.5) and 1% DTT (ditiotreitol), under nitrogen. The determination was carried out using a 1050 series apparatus Hewlett Packard (Palo Alto, CA, USA) with the following parameters: a RP-18 Vydac 218TP54 column with 5 μm bead size and 300 A pore size, 250 × 4.6 mm; a Zorbax 300SB C18 pre-column, 4.6 × 12.5 mm; a column temperature of 45 °C, a mobile phase flow rate of 1 mL $min^{-1}$ and an injection volume of 20 μL.

A two-component gradient was used. Component A: 0 min 75%, 5 min 65%, 10 min 50%, 17 min 25%, 18 min 15% and 19 min 75%. The first component (A) was water with 0.1% of trifluoroacetic acid (TFA) and the second (B) was acetonitrile (ACN) with 0.1% of TFA. The absorbance spectra of eluted proteins were determined by a diode-array detector HP 1050 (Palo Alto, CA, USA). Quantification of proteins was done by UV absorbance at 210 nm. The identification of protein subunits was based on their retention times and the second derivative of their UV spectra according to Konopka et al. [37]. The results were analysed with the use of the computer program HPLC 3D ChemStation (Palo Alto, CA, USA) and presented in mAU $s^{-1}$ (milli-absorbance units).

### 2.6. Statistical Analysis

The obtained results were statistically analysed with the analysis of variance (ANOVA). Tukey's test was used to determine a statistically significant difference at the level of $p = 0.05$. Statistical analysis of the results was performed using the TIBCO Statistica 13.3.0 (TIBCO Software Inc., Palo Alto, CA, USA).

## 3. Results and Discussion

### 3.1. Physiological Parameters

The leaf area index (LAI), which determines the coverage of the canopy, is the most important parameter reflecting the dynamic growth rate of crops. The LAI determines the size of the area of photosynthetic active radiation (PAR) on which the effectiveness of the photosynthesis process depends [38]. The LAI achieved the highest value in the CT system compared to other tillage systems, by 10.3% (RT) and 8.3% (NT), respectively (Table 4). A similar relationship in studies with spring wheat was obtained by Kulig et al. [39], where this ratio was higher by an average of 12.0% for facilities in the CT system compared to RT. Liu et al. [40] also showed that, regardless of the wheat cultivar tested, a higher LAI value was obtained in the CT system compared to NT. Cv. Hypocamp obtained a higher LAI value compared to cvs. Hybery (by 9.2%), Hymalaya (by 9.7%) and Hyvento (by 4.7%).

A statistically significant interaction between the experimental factors in shaping the value of this parameter was found. The highest value of LAI was achieved by cvs. Hypocamp (6.1) and Hyvento (6.0) in the CT system and Hyking (5.9) in the NT system (Figure 2a).

The content of chlorophyll is an important indicator that directly influences the absorption, transmission and distribution of light energy and the efficiency of the photosynthesis process [17]. The chlorophyll content in the flag leaf was significantly the highest in the CT system (31.7 chlorophyll content index (CCI) (Table 4)). Compared to RT and NT systems, the difference was 4.3 and 9.3%, respectively. The studies by Liu and Wiatrak [41] did not show a significant effect of different tillage systems on the content of chlorophyll in maize leaves, but an increase in this parameter was found in the season with the highest rainfall. Cv. Hypocamp, compared to the other cultivars, was characterised by the highest value of this parameter (32.8 CCI), while cv. Hyvento (28.4 CCI) the lowest. The highest chlorophyll content was found for cv. Hypocamp in the CT and RT systems and for cv. Hybery in CT, and the lowest for cv. Hyvento in RT and NT (Figure 2b).

Chlorophyll fluorescence is electromagnetic radiation emitted by chlorophyll in plants. The analysis of chlorophyll fluorescence shows the ability of the photosynthetic apparatus to collect light energy and use it in electron transport in photosynthesis [42]. Chlorophyll fluorescence measurements are therefore used to identify drought-tolerant cereal genotypes in which susceptible cultivars are characterised by reduced PSII connectivity and inhibition of electron supply from the water-splitting system (sensitive lines were indicated by decreasing PSII connectivity and inhibiting electron supply from the water-splitting system, respectively) [43]. The tested cultivars did not differentiate the value of the PI parameter. Among the cultivars, the highest values of chlorophyll fluorescence parameters were achieved by cv. Hypocamp, but only for the PI parameter it was not a statistically significant difference. Cv. Hyking had the lowest $F_v/F_m$ value, while cv. Hyvento the lowest $F_v/F_0$. Selected parameters of chlorophyll fluorescence ($F_v/F_m$, $F_v/F_0$ and PI) in the CT system achieved a higher value compared to NT and RT (Table 4, Figure 2c–e).

The highest values of LAI parameters, chlorophyll content and $F_v/F_0$ were obtained in the 2017/2018 season. In the case of the $F_v/F_m$ and PI parameters, the values were high in both the 2017/2018 and 2016/2017 seasons. The research conducted by Stępień-Warda [44] on maize plants cultivated in a temperate climate showed that the tillage system (CT, RT and NT) combined with the year of research (in combination with weather conditions) had a significant impact on the course of the photosynthesis process. In our own study in conditions of insufficient moisture content (2018/2019 season), higher values of $F_v/F_m$ and PI parameters were achieved in the RT system compared to CT. In addition, Badr and Brüggemann [43] observed a reduction of $F_v/F_m$ and PI parameters in maize plants as a result of drought stress. Hou et al. [17] showed that under conditions of low water level caused by drought, wheat plants grown in reduced tillage systems (RT, NT) synthesised chlorophyll better, which contributed to the improvement of photosynthesis in later growth stages compared to the conventional system (CT).

Gas exchange, apart from the analysis of chlorophyll a fluorescence, is also a valuable, non-invasive method of assessing the condition of plants [45]. Gas exchange parameters ($g_s$ and E) were the highest in the CT system, while in the case of $P_N$, they did not differ significantly from the RT system (Table 4). The $C_i$ parameter was not differentiated by tillage systems, but the highest 215.9 µmol $CO_2$ m$^{-2}$ s$^{-1}$ was recorded in the CT system. The values of the $P_N$ parameter in the studied cultivars ranged from 18.5 to 19.5 µmol m$^{-2}$ s$^{-1}$. Cv. Hyvento achieved the lowest $P_N$ values compared to the other cultivars. For $g_s$, no significant varietal differences were found, while for parameter E, cv. Hyvento obtained higher values compared to cvs. Hyking and Hymalaya, by 9.1 and 10.4%, respectively. The values of the $C_i$ parameters for cvs. Hymalaya and Hypocamp were higher than for cvs. Hybery and Hyking. Skider et al. [46] showed that water use efficiency (WUE) determined on the basis of the quotient of net photosynthesis intensity to transpiration ($P_N/E$) is an important physiological parameter that indicates the possibility of growing plants in regions with low water availability. WUE is a key indicator in the selection of genotypes cultivated in arid and

semi-arid areas. The WUE index achieved the lowest value in the CT system compared to RT and NT (by 4.2 and 6.0%). WUE achieves higher values on soils cultivated in the RT system than in the CT system. Compared to the CT system, the NT system affects the water-saving capacity and alleviates drought stress during the growing season in dry conditions [47]. In drought conditions, the NT system has an advantage over CT due to the increased availability of water, which increases photosynthetic activity, resulting in an increase in yield [48]. Cvs. Hyking and Hymalaya were characterised by a higher value of this parameter compared to cv. Hyvento by 15.7 and 18.0%, respectively.

For all gas exchange parameters, statistically significant interactions between the tillage systems and cultivars were found. The highest values of $g_s$ and E were obtained by cvs. Hymalaya, Hypocamp and Hyvento and $P_N$ by cvs. Hymalaya and Hypocamp in the CT system (Figure 2f–h). However, in RT, and in particular NT, these cultivars were characterised by the lowest values of these parameters. In contrast, cvs. Hybery and Hyking responded better to cultivation in RT and NT systems than in CT systems. The $C_i$ parameter reached the highest value for cvs. Hymalaya and Hypocamp in NT compared to cvs. Hyvento in CT, Hyking in RT and NT, and Hybery in RT and NT (Figure 2i). The WUE index, on the other hand, achieved the highest value for cv. Hymalaya in NT and the lowest for cvs. Hymalaya, Hypocamp and Hyvento in CT (Figure 2j).

The gas exchange parameters ($P_N$, $g_s$ and E) in the 2018/2019 growing season reached the lowest values, while the $C_i$ parameter and the WUE index were the highest. Earlier studies show that water stress may reduce the photosynthetic rate of plants [49]. Water scarcity is an important factor limiting the photosynthesis process in agricultural crops [17]. Inhibition of photosynthesis due to drought stress reduces plant growth and yielding [50]. According to Gao et al. [51], a decrease in soil water content decreased the values of $P_N$ and E (net photosynthetic rate and transpiration rate). Conservation crops alleviate the stress of drought in the later growing season of wheat, which leads to the enhancement of photosynthetic properties and increases the grain yield [17].

*3.2. Grain Yield*

The grain yield was significantly dependent on the tillage systems, cultivar and year of research. The highest grain yield was found in CT in comparison to RT (by 4.5%) and NT (by 3.5%) (Table 4). Research by Małecka et al. [52] confirms an increase in the yield of barley grain by 6.8% in the CT system in comparison with the NT system, while the yield obtained in NT was close to RT. In studies by Šíp et al. [24], the yield of wheat grain was dependent to a greater extent on soil moisture and the cultivar genotype, and the difference in yield between the CT and RT systems was, similarly to the authors' own study, 3.0%.

Cv. Hypocamp was characterised by a higher grain yield compared to cvs. Hybery, Hymalaya and Hyvento; however, cvs. Hymalaya and Hyvento yielded the lowest (Figure 3a). In the CT system, the highest grain yield was obtained by cvs. Hypocamp (9.72 t ha$^{-1}$) and Hyvento (9.43 t ha$^{-1}$), and in the NT cv. Hyking (9.58 t ha$^{-1}$). The lowest grain yield was found for cvs. Hyvento (8.29 t ha$^{-1}$ in RT and 8.03 t ha$^{-1}$ in NT) and Hymalaya (8.30 t ha$^{-1}$ in RT and 8.55 t ha$^{-1}$ in NT). Cv. Hybery obtained grain yield at a similar level in all tillage systems. However, in the case of cv. Hyking, a 9.7% higher grain yield was found in NT compared to CT.

**Table 4.** Selected physiological parameters.

| Factor | | Grain Yield $(t\ ha^{-1})$ | LAI | Relative Chlorophyll Content (CCI) | Chlorophyll Fluorescence | | | Gas Exchange | | | | |
|---|---|---|---|---|---|---|---|---|---|---|---|---|
| Tillage (T) | Cultivar (C) | | | | $F_v/F_m$ | $F_v/F_0$ | PI | $P_N$ $(\mu mol\ CO_2$ $m^{-2}\ s^{-1})$ | $g_s$ $(mol\ H_2O$ $m^{-2}\ s^{-1})$ | E $(mmol\ H_2O$ $m^{-2}\ s^{-1})$ | $C_i$ $(mmol\ L^{-1})$ | WUE $(mmol \cdot mol^{-1})$ |
| CT | | 9.13 ᵇ | 5.38 ᵇ | 31.7 ᶜ | 0.797 ᵇ | 4.11 ᵇ | 5.38 ᵇ | 19.3 ᵇ | 0.357 ᵇ | 4.35 ᵇ | 215.9 ᵃ | 4.54 ᵃ |
| RT | | 8.74 ᵃ | 4.88 ᵃ | 30.4 ᵇ | 0.790 ᵃ | 3.96 ᵃ | 4.88 ᵃ | 19.1 ᵇ | 0.322 ᵃ | 4.07 ᵃ | 217.0 ᵃ | 4.73 ᵇ |
| NT | | 8.82 ᵃ | 4.97 ᵃ | 29.0 ᵃ | 0.793 ᵃ | 3.93 ᵃ | 4.97 ᵃ | 18.9 ᵃ | 0.333 ᵃ | 3.98 ᵃ | 217.4 ᵃ | 4.81 ᵇ |
| | Hybery | 8.83 ᵃᵇ | 4.88 ᵃ | 31.1 ᶜ | 0.789 ᶜ | 4.07 ᵇ | 4.88 ᵃ | 19.1 ᵇ | 0.335 ᵃ | 4.16 ᵃᵇ | 211.9 ᵃ | 4.63 ᵃᵇ |
| | Hyking | 9.12 ᵇᶜ | 5.22 ᵃᵇ | 30.0 ᵇᶜ | 0.779 ᵃ | 4.02 ᵇ | 5.22 ᵃᵇ | 19.5 ᵇ | 0.333 ᵃ | 3.98 ᵃ | 212.7 ᵃ | 4.95 ᵇ |
| | Hymalaya | 8.62 ᵃ | 4.86 ᵃ | 29.6 ᵇ | 0.802 ᶜ | 4.05 ᵇ | 4.86 ᵃ | 19.3 ᵇ | 0.340 ᵃ | 3.93 ᵃ | 221.4 ᵇ | 5.05 ᵇ |
| | Hypocamp | 9.33 ᶜ | 5.33 ᵇ | 32.8 ᵈ | 0.800 ᶜ | 4.09 ᵇ | 5.33 ᵇ | 19.1 ᵇ | 0.340 ᵃ | 4.27 ᵃᵇ | 220.4 ᵇ | 4.55 ᵃᵇ |
| | Hyvento | 8.58 ᵃ | 5.09 ᵃ | 28.4 ᵃ | 0.797 ᵇᶜ | 3.78 ᵃ | 5.09 ᵃ | 18.5 ᵃ | 0.337 ᵃ | 4.34 ᵇ | 217.4 ᵃᵇ | 4.28 ᵃ |
| Year (Y) | | | | | | | | | | | | |
| 2016/2017 | | 9.06 ᵇ | 5.23 ᵇ | 31.2 ᵇ | 0.791 ᵃᵇ | 4.08 ᵇ | 8.07 ᵃᵇ | 19.3 ᵃᵇ | 0.343 ᵇ | 4.27 ᵇ | 217.8 ᵇ | 4.54 ᵃ |
| 2017/2018 | | 9.67 ᶜ | 5.65 ᶜ | 33.1 ᶜ | 0.804 ᵇ | 4.18 ᶜ | 8.15 ᵇ | 19.6 ᵇ | 0.348 ᵇ | 4.24 ᵇ | 212.8 ᵃ | 4.64 ᵃᵇ |
| 2018/2019 | | 7.96 ᵃ | 4.36 ᵃ | 26.9 ᵃ | 0.783 ᵃ | 3.75 ᵃ | 7.70 ᵃ | 18.5 ᵃ | 0.319 ᵃ | 3.88 ᵃ | 219.7 ᵇ | 4.79 ᵇ |
| Mean | | 8.90 | 5.08 | 30.4 | 0.793 | 4.00 | 7.97 | 19.1 | 0.337 | 4.13 | 216.8 | 4.69 |
| T | | *** | * | *** | *** | *** | *** | *** | *** | *** | ns | ** |
| C | | *** | *** | *** | *** | *** | * | * | ns | * | * | *** |
| Y | | *** | *** | *** | ** | *** | * | * | * | *** | ** | ** |
| T × C | | *** | *** | *** | *** | *** | *** | *** | *** | *** | *** | *** |
| T × Y | | *** | * | *** | *** | * | ** | * | * | *** | *** | *** |
| C × Y | | *** | ns | *** | *** | *** | ** | *** | *** | *** | * | *** |
| T × C × Y | | ** | ns | *** | *** | *** | *** | *** | *** | *** | *** | *** |

Mean values marked with different letters in the same column indicate significant differences ($p = 0.05$), according to ANOVA followed by Tukey's test. *, **, *** and ns mean $\leq 0.05$, $\leq 0.01$, $< 0.001$, and not significant, respectively. LAI: leaf area index, CCI: chlorophyll content index, $F_v/F_m$: maximal quantum yield of PSII photochemistry, $F_v/F_0$: the maximum quantum yield of primary photochemistry, PI: performance index, $P_N$: Net photosynthetic rate, $g_s$: stomatal conductance, E: transpiration rate, $C_i$: intracellular $CO_2$ concentration, WUE: water use efficiency

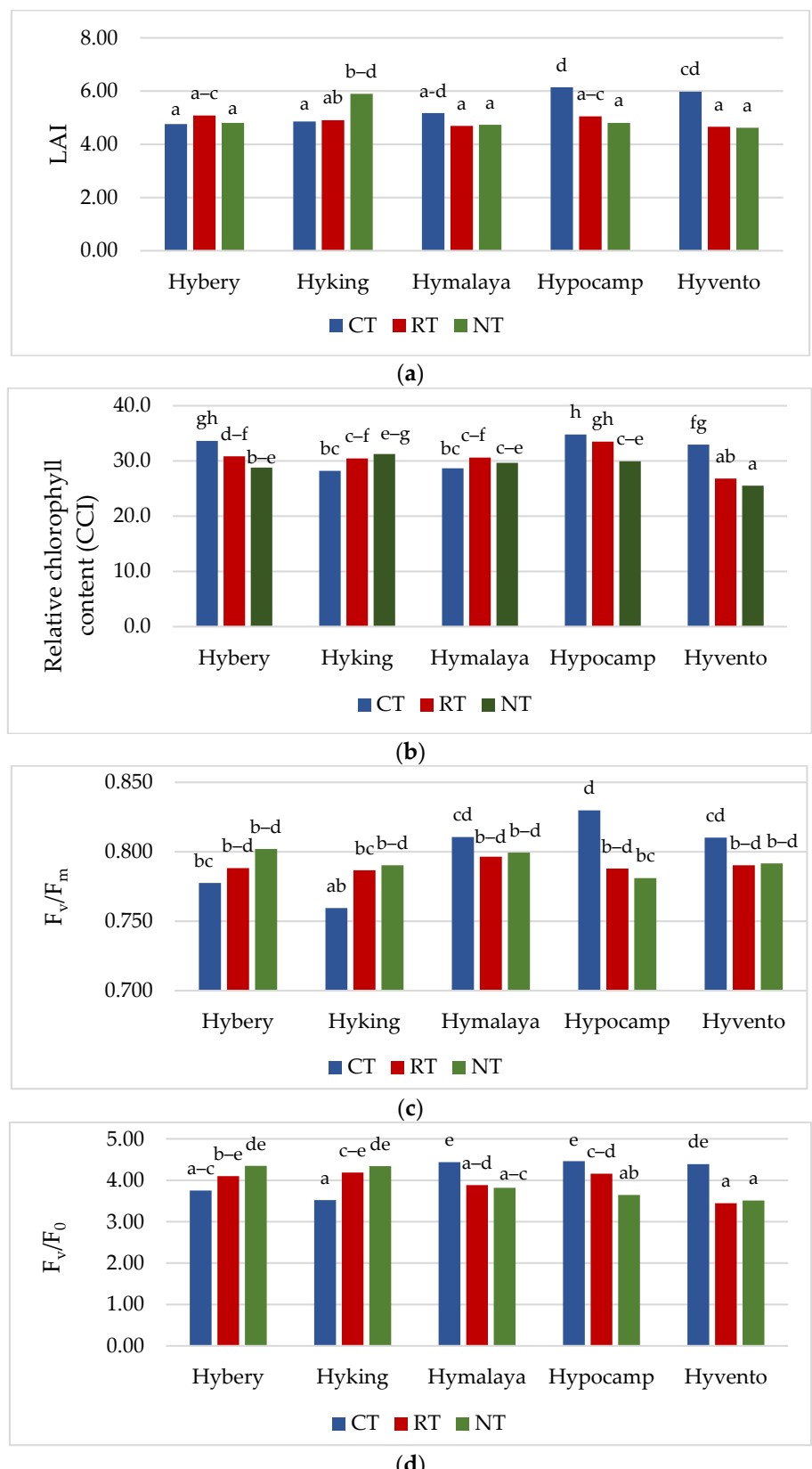

**Figure 2.** *Cont.*

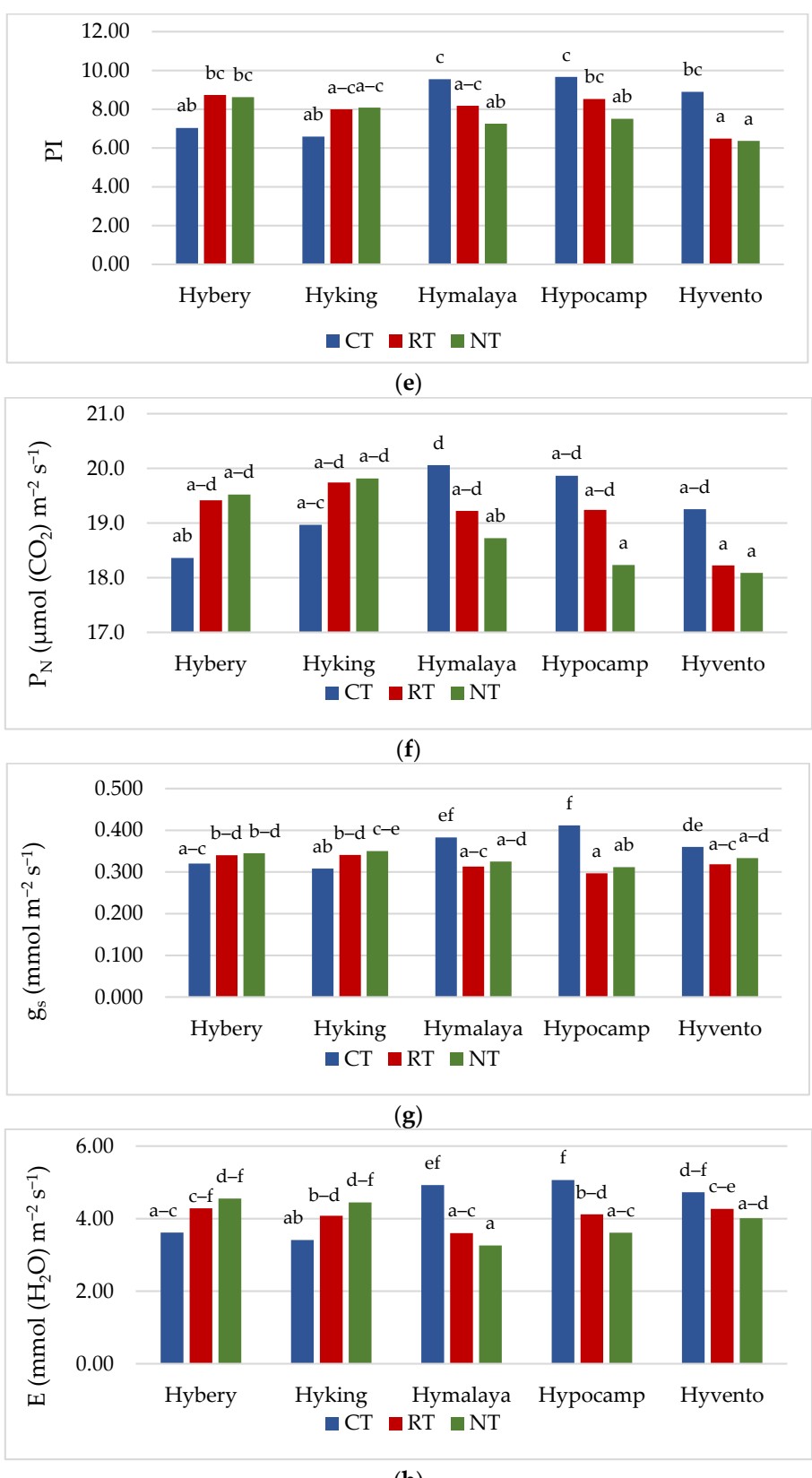

**Figure 2.** *Cont.*

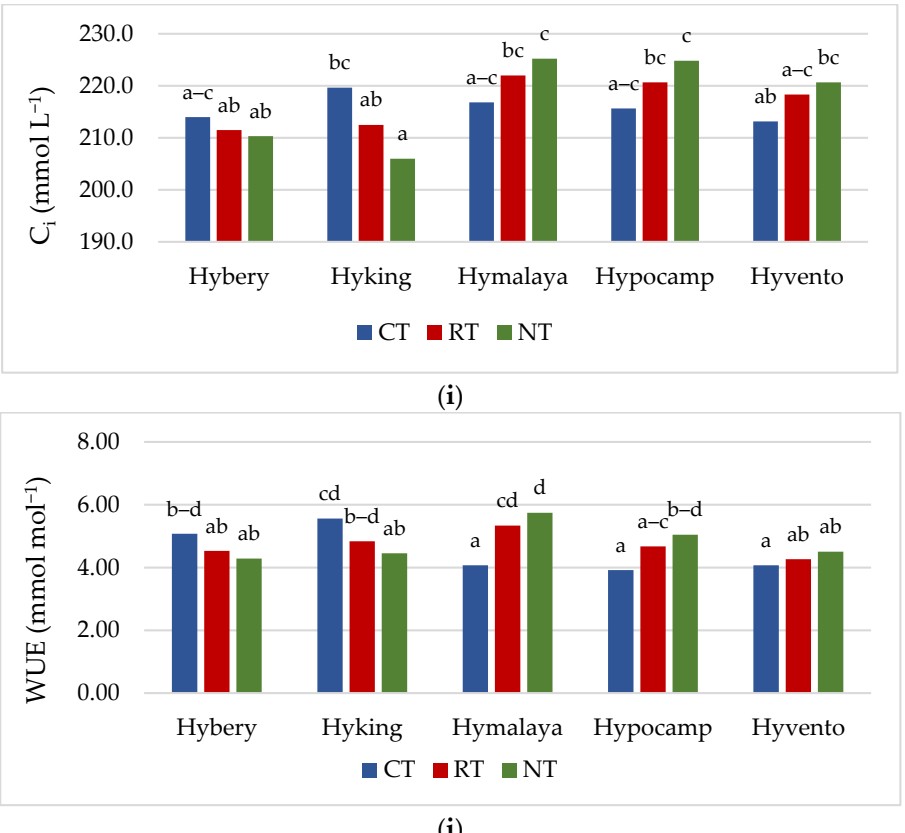

**Figure 2.** (**a–e**) Effect of tillage systems and hybrid winter wheat cvs. on LAI, relative chlorophyll content and selected chlorophyll fluorescence parameters. Different letters indicate significant differences according to ANOVA (followed by Tukey's HSD test, *p* = 0.05). LAI: leaf area index, CCI: chlorophyll content index, $F_v/F_m$: maximal quantum yield of PSII photochemistry, $F_v/F_0$: the maximum quantum yield of primary photochemistry, PI: performance index. (**f–j**). Effect of tillage systems and hybrid winter wheat cvs. on selected gas exchange parameters. Different letters indicate significant differences according to ANOVA (followed by Tukey's HSD test, *p* = 0.05). $P_N$: Net photosynthetic rate, $g_s$: stomatal conductance, E: transpiration rate, Ci: intracellular $CO_2$ concentration, WUE: water use efficiency.

The highest grain yield, 10.24 t ha$^{-1}$, was obtained in CT in the years 2017/2018, when the most favourable hydrothermal conditions supported the development of wheat plants (Figure 3b). In the 2016/2017 season, the grain yield in CT was higher than in RT (by 10.0%) and NT (by 13.0%). In the case of the 2018/2019 season, where the values of the Sielianinov coefficient were low, the highest grain yield (8.66 t ha$^{-1}$) was obtained in NT. Similar relationships were found in the studies conducted by Cociu and Alionte [53] and Woźniak and Rachoń [54], where the lowest yield of winter wheat grain cultivated in the CT system was obtained in the season characterised by a shortage of rainfall.

All tested cultivars of hybrid winter wheat yielded the best in the 2017/2018 season (Figure 3c). Significant differences between the years of research in the grain yield occurred only in cv. Hymalaya, which in the 2017/2018 season was characterised by a grain yield higher by 11.1 and 25.8% compared to the 2016/2017 and 2018/2019 seasons. In the case of the other cvs. Hybery, Hyking, Hypocamp and Hyvento, no significant differences were found in the grain yields in 2016/2017 and 2017/2018.

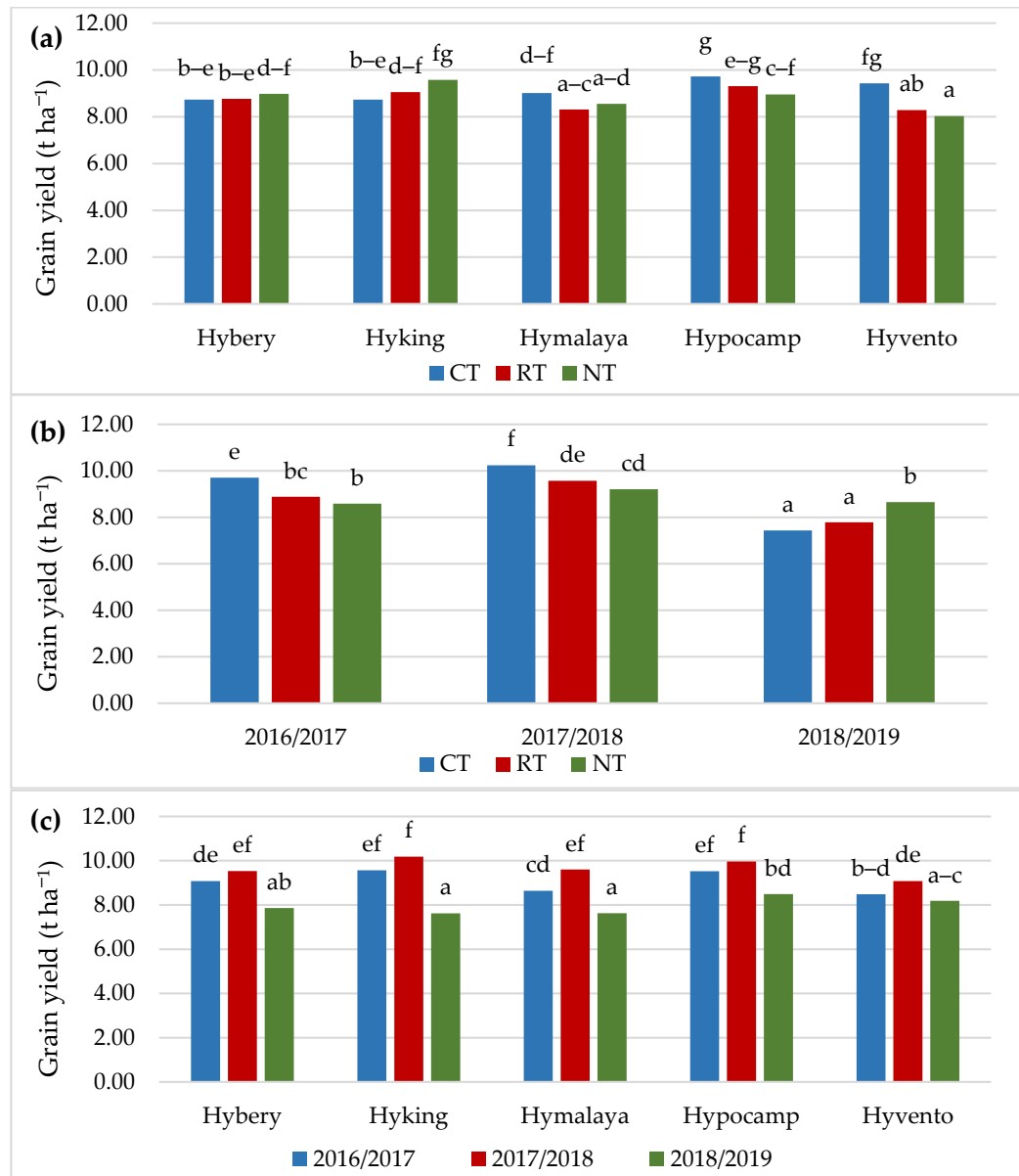

**Figure 3.** Grain yield depending on (**a**) cultivars and tillage, (**b**) tillage and years of research, (**c**) cultivars and years of research. Different letters indicate significant differences according to ANOVA (followed by Tukey's HSD test, $p = 0.05$).

The tillage system largely influences the growth and differentiation of the elements of the yield structure that determine the size of the obtained grain yield [55]. According to research by Rieger et al. [56] a reduction in grain yield as a result of reduction in tillage was the result of lower ear density and TGW (thousand grain weight). In a study by Jaskulska [57], the interaction effect of the use of different systems of cultivation of winter wheat and the year of research was the result of the different effects of weather conditions on soil properties. In reduced tillage or no-tillage systems (RT, NT), higher grain yield may be the result of different physico-chemical properties of the soil due to the accumulation of organic matter and the formation of soil aggregates [58]. The reason for this may be the positive effect of zero tillage (NT) on the water properties of the soil when the evaporation of water from the field surface is limited. In soils characterised by moderate moisture content, such conditions are created in the CT system, while in arid and semi-arid soils, the NT system provides better conditions [59,60]. In addition, the presence of stubble remains in the NT system reduces evaporation and a lower number of macropores and a higher volume of medium-sized water holding pores also cause higher water content in the soil as

a result of the use of conservation tillage [52]. Compared to the CT system, the NT system, by avoiding disturbing the soil surface structure, improves its structure, and by covering with mulch, it promotes better water infiltration and reduces its loss from the soil through evaporation and increases the content of organic matter in the soil [61].

### 3.3. Quality Parameters

A significantly higher content of protein in the grain as well as gluten, sedimentation index, falling number and flour gluten index were obtained in the CT system compared to NT (Table 5). Protein and gluten contents did not differ statistically between the CT and RT systems, and the falling number values were similar for RT and NT. According to the research of Woźniak and Gos [62] and Taner et al. [22], soil tillage systems did not significantly differentiate the basic quality characteristics of wheat. According to Šip et al. [24], the CT system compared to RT and NT results in better nitrogen utilisation, which influences growth, especially of the protein content in wheat grain. In the studies by Jaskulska et al. [21], similarly to our own study, a reduction was shown in ash content in flour from the CT system compared to RT and NT.

**Table 5.** Selected qualitative indicators depending on experimental factors.

| Factor | | Crude Protein (g kg⁻¹) | Wet Gluten (%) | Zeleny Test (mL) | Falling Number (s) | Total Ash (%) | Gluten Index (%) |
|---|---|---|---|---|---|---|---|
| Tillage (T) | Cultivar (C) | | | | | | |
| CT | | 130 [b] | 27.6 [b] | 46.1 [c] | 315 [b] | 0.48 [a] | 97 [c] |
| RT | | 127 [b] | 26.2 [b] | 38.8 [b] | 293 [a] | 0.51 [b] | 95 [b] |
| NT | | 117 [a] | 22.6 [a] | 34.8 [a] | 300 [a] | 0.54 [b] | 91 [a] |
| | Hybery | 123 [a] | 25.4 [b] | 37.3 [a] | 288 [b] | 0.48 [a] | 97 [d] |
| | Hyking | 126 [b] | 26.0 [b] | 43.2 [b] | 319 [c] | 0.57 [c] | 94 [bc] |
| | Hymalaya | 121 [a] | 24.0 [a] | 35.5 [a] | 335 [d] | 0.51 [ab] | 96 [cd] |
| | Hypocamp | 123 [a] | 24.1 [a] | 37.3 [a] | 252 [a] | 0.47 [a] | 92 [ab] |
| | Hyvento | 132 [c] | 27.9 [c] | 46.0 [c] | 318 [c] | 0.53 [b] | 92 [a] |
| Year (Y) | | | | | | | |
| 2016/2017 | | 117 [a] | 23.7 [a] | 35.5 [a] | 286 [a] | 0.50 [a] | 94 [a] |
| 2017/2018 | | 125 [b] | 25.5 [b] | 40.6 [c] | 289 [a] | 0.50 [a] | 95 [a] |
| 2018/2019 | | 132 [c] | 27.3 [c] | 43.5 [b] | 333 [b] | 0.54 [a] | 93 [a] |
| Mean | | 125 | 25.3 | 39.9 | 302 | 0.51 | 94 |
| T | | *** | *** | *** | *** | *** | *** |
| C | | *** | *** | *** | *** | *** | *** |
| Y | | *** | *** | *** | *** | ns | ns |
| T × C | | ** | ns | ns | ns | *** | *** |
| T × Y | | ** | ns | * | ns | ns | ns |
| C × Y | | *** | ** | *** | *** | *** | ns |
| T × C × Y | | ns | ns | ns | ns | ns | ns |

Mean values marked with the different letters in the same column indicate significant differences (*p* = 0.05), according to ANOVA followed by Tukey's test. *, **, *** and ns mean ≤0.05, ≤0.01, <0.001, and not significant, respectively.

According to Woźniak and Rachoń [54], the combination of genetic features of wheat cultivars and environmental conditions as well as their interaction have a greater impact than tillage systems on the quality characteristics of wheat grain and flour. Research by Linina and Ruža [63] shows that the values of wheat quality parameters, including the value of the falling number, significantly depended on the weather conditions, grain storage period and the nitrogen dose applied. In the authors' own study, the 2018/2019 growing season with a dry ripening period of wheat in June and July was favourable for a significantly higher value of the quality parameters of wheat grain and flour [21,54]. There was no statistical variation between the study years for the ash content and the gluten index.

According to many authors [64–66], weather conditions with excess rainfall may affect the faster synthesis of gliadins in protein, thus weakening the mechanical strength of gluten, which is a feature of the wheat cultivar genotype.

The protein and gluten content in the tested wheat cultivars ranged from 121 to 132 g kg⁻¹ and from 24.0 to 27.9%, respectively, and the sedimentation index range from 35.5 to 46.0 mL.



Cvs. Hyvento and Hyking from the CT system were distinguished by the clearly highest content of both protein in grain as well as gluten and sedimentation index in flour. Among the cultivars, the lowest values of the above-mentioned quality parameters in the NT system were shown by cv. Hymalaya.

The content of the above-mentioned quality parameters of the hybrid winter wheat cultivars studied did not differ significantly from the values given in the studies by Šip et al. [24] and Jaskulska et al. [21]. According to Bueno et al. [67], most of the flours intended for the baking industry require the falling number to be adjusted to the ideal ranges from 250 to 320 s and the reduction of this parameter value is achieved by adding an α-amylase preparation.

In our study, this condition is fulfilled by flour of wheat cultivars from all tillage systems, with the exception of cv. Hymalaya. The smallest value falling number of 238 (RT) to 272 s (CT) was noted for cv. Hypocamp. With the exception of cv. Hyking, other cultivars were characterised by a significantly lower total ash content in the flour in the CT system compared to the NT system, ranging from 12.7 (cv. Hymalaya) to 21.2% (cv. Hypocamp). The reduction of ash content in grain in the CT system compared to NT was from 1.4 to 8.6% in the studies by Jaskulska et al. [21] and Woźniak and Rachoń [54]. The value of the gluten index in the flour of most hybrid cultivars ranged from 92 to 97. According to Ćurić et al. [68], a flour with a gluten index in the range of 75 to 90 usually provides optimal baking quality. Thus, flour from cvs. Hypocamp, Hyvento and Hyking was rated more favourably especially when grown in the NT system, and had a gluten index ranging from 86 to 91. The research of Šip et al. [24] shows that the gluten index of wheat cultivars ranged widely from 65 to 98. This index tended to be higher in flour of wheat cultivars in the RT system than in CT, as opposed to our own study, where the gluten index was very low in the NT system compared to CT and RT.

Protein Composition and Characterisation

The protein fractional composition depended significantly on the tillage systems, cultivars, the year of research and interactions between experimental factors (Table 6).

**Table 6.** Composition and ratio protein fractions dependrting on experimental factors.

| Factor | | Albumins Globulins | Gliadins (Gli) | | | | Glutenins (Glu) | | | Ratio | |
|---|---|---|---|---|---|---|---|---|---|---|---|
| | | | α/β | γ | ω | Σ Gli [1] | H [3] | L [4] | Σ Glu [2] | | |
| Tillage (T) | Cultivar (C) | | mAU·s$^{-1}$ | | | | | | | H/L [5] | Gli/Glu [6] |
| CT | | 11.7 [b] | 15.6 [b] | 9.4 [c] | 4.9 [c] | 29.9 [c] | 5.6 [c] | 18.8 [b] | 24.4 [c] | 0.30 [a] | 1.23 [a] |
| RT | | 11.0 [b] | 14.2 [b] | 8.1 [b] | 4.0 [b] | 26.3 [b] | 4.7 [b] | 15.8 [b] | 20.6 [b] | 0.31 [a] | 1.30 [a] |
| NT | | 9.30 [a] | 12.4 [a] | 7.3 [a] | 3.2 [a] | 23.0 [a] | 4.0 [a] | 13.3 [a] | 17.3 [a] | 0.31 [a] | 1.36 [a] |
| | Hybery | 10.7 [b] | 13.6 [a] | 7.8 [a] | 3.6 [b] | 25.0 [b] | 4.3 [b] | 14.8 [a] | 19.2 [a] | 0.30 [ab] | 1.33 [b] |
| | Hyking | 11.5 [c] | 13.6 [a] | 8.5 [b] | 4.1 [c] | 26.2 [c] | 4.8 [c] | 17.6 [b] | 22.4 [b] | 0.28 [a] | 1.18 [a] |
| | Hymalaya | 9.8 [a] | 13.1 [a] | 7.3 [a] | 3.3 [a] | 23.6 [a] | 4.0 [a] | 14.3 [a] | 18.3 [a] | 0.29 [a] | 1.31 [b] |
| | Hypocamp | 10.0 [a] | 14.7 [b] | 8.8 [b] | 3.7 [b] | 27.1 [c] | 4.4 [b] | 14.3 [a] | 18.8 [a] | 0.33 [c] | 1.48 [c] |
| | Hyvento | 11.4 [c] | 15.4 [c] | 8.9 [b] | 5.5 [d] | 29.9 [d] | 6.3 [d] | 18.9 [c] | 25.1 [c] | 0.33 [bc] | 1.19 [a] |
| Year (Y) | | | | | | | | | | | |
| 2016/2017 | | 9.7 [a] | 12.7 [a] | 6.9 [a] | 3.5 [a] | 23.1 [a] | 4.2 [a] | 13.5 [a] | 17.7 [a] | 0.31 [a] | 1.33 [a] |
| 2017/2018 | | 11.1 [b] | 14.5 [b] | 8.5 [b] | 4.0 [b] | 27.0 [b] | 4.7 [b] | 16.2 [b] | 20.9 [b] | 0.30 [a] | 1.32 [a] |
| 2018/2019 | | 11.3 [b] | 15.0 [c] | 9.4 [b] | 4.7 [c] | 29.0 [c] | 5.4 [c] | 18.2 [b] | 23.6 [c] | 0.30 [a] | 1.25 [a] |
| Mean | | 10.7 | 14.1 | 8.3 | 4.1 | 26.4 | 4.7 | 16.0 | 20.7 | 0.30 | 1.30 |
| T | | *** | *** | *** | *** | *** | *** | *** | *** | ns | ns |
| C | | *** | *** | *** | *** | *** | *** | *** | *** | *** | *** |
| Y | | *** | *** | *** | *** | *** | *** | *** | *** | ns | ns |
| T × C | | *** | * | ** | *** | ** | *** | ** | * | *** | * |
| T × Y | | ** | ns | ns | ns | *** | * | ns | * | ns | ns |
| C × Y | | *** | *** | *** | *** | *** | *** | *** | *** | *** | * |
| T × C × Y | | ns | ns | ns | ns | ns | ns | * | ns | ns | * |

Mean values marked with the different letters in the same column indicate significant differences (*p* = 0.05), according to ANOVA followed by Tukey's test. *, **, *** and ns mean ≤0.05, ≤0.01, <0.001, and not significant, respectively. [1] Σ Gli: sum gliadins; [2] Σ Glu: sum glutenins; [3] H: HMW—high molecular weight; [4] L: LMW—low molecular weight; H/L[5]: high molecular weight/low molecular weight; Gli/Glu [6]: gliadins/glutenins.

A significant increase was found in the sum of gliadins and glutenins in grain from the CT system in comparison to RT and NT. The increase in the amount of gliadins from the CT system in relation to RT and NT was 12.0 and 23.1% and for glutenin it was 15.6 and 29.1%. There were no significant differences in the content of albumin and globulin in the RT and NT systems. In addition, Pagnani et al. [69] showed a higher amount of gliadins and glutenins from the CT system in hard wheat grain compared to NT, especially when field beans were the forecrop of wheat, which was conducive to higher N-remobilisation to the grains. According to Garcia-Molina and Barro [65], the increased content of gliadins and glutenins in wheat grain is mainly determined by increasing nitrogen doses and the genotype of the cultivar. According to Horvat et al. [19], albumin and globulins account for 12.2 to 19.8% of the total wheat flour protein while gliadins and glutenins account for 80.3 to 87.8%.

In our study in relation to general protein share, the share of storage proteins (gliadin and glutenin) definitely prevailed (from 81.0 and 81.3% in RT and NT to 82.3% in CT) compared to enzymatic and structural proteins (albumin and globulins).

The use of the CT system compared to RT and NT resulted in a significant increase in the content of $\gamma$ and $\omega$ gliadins and HMW glutenins. No statistical differences were found in the content of $\alpha/\beta$ gliadins and LMW glutenins in the CT and RT systems. In the gliadin fraction of hybrid winter wheat grains, the $\alpha/\beta$, $\gamma$ and $\omega$ gliadins constituted on average 53.3, 31.3 and 15.4%. According to Shewry [2], the content of the HMW and LMW glutenin fractions is important in determining the quality and end use of grain. In our study, HMW protein subunits accounted for 22.7% and LMW for 77.3% in the total content of polymeric glutenins. In the studies by Ramírez-Wong et al. [70] clearly the highest number of $\alpha/\beta$, $\gamma$ and $\omega$ gliadins and HMW and LMW glutenin were found in wheat after the application of a dose of 180 kg N ha$^{-1}$ compared to a dose of 60 kg N ha$^{-1}$. Buczek et al. [71] showed an increase in gliadin and glutenin subunits in grains from high input technology with higher NPK fertilisation and the use of pesticides compared to low input technology.

According to Hofmeijer et al. [25] lower soil compaction in the CT system compared to RT and NT may cause higher uptake of nutrients, including nitrogen, by wheat, which results in higher grain quality, mainly when the weather conditions for the mineralisation of post-harvest residues in RT and NT are unfavourable.

In our own study, interactions between tillage systems and cultivars, and between cultivars and year of research, indicate a significant influence of environmental conditions on the content of gliadins and glutenins and their subunits [19,72]. A more favourable composition of proteins and their subunits in the seed of the cultivars was found in the 2018/2019 season, where the sum of precipitation was lower by 186.5 mm, and the average air temperature was higher by 1.6 °C than the average over many years. Only cv. Hyvento did not show any variability in the content of albumin and globulin in the study years and the sum of gliadins in the 2017/2018 and 2018/2019 growing seasons (Figures 4b and 5b).

The grain of the cultivars from the CT system, compared to RT and NT, contained significantly more $\alpha/\beta$, $\gamma$ subunits (except for cvs. Hybery and Hymalaya) and $\omega$ gliadins as well as HMW and LMW glutenins, and also the sums of gliadins and glutenins (Figures 4a and 5a).

From the examined cultivars, the highest values of the above-mentioned gluten protein fractions in the CT system were found in the grain of cv. Hyvento and the lowest in cv. Hymalaya in the NT. On average cvs. Hyking and Hyvento accumulated significantly more albumin and globulin in the grain and cvs. Hyking, Hypocamp and Hyvento–$\gamma$ gliadyn.

In addition, Horvat et al. [19] showed environmental and genotypic (varietal) variability of gliadin and glutenin subunits in wheat grain, and the Gli/Glu ratio of the analysed cultivars ranged from 1.35 to 2.09. According to Peigné et al. [20] a higher Gli/Glu ratio may suggest reduced technological quality of wheat cultivar protein. In the authors' own study, the Gli/Glu and H/L ratios of the analysed cultivars ranged from 1.18 to 1.48 and from 0.28 to 0.33. The Gli/Glu and H/L ratios were not significantly differentiated by the tillage systems and the year of research, and their more favourable values were shown respectively by cvs. Hyking and Hyvento and cvs. Hyvento and Hypocamp (Table 6).

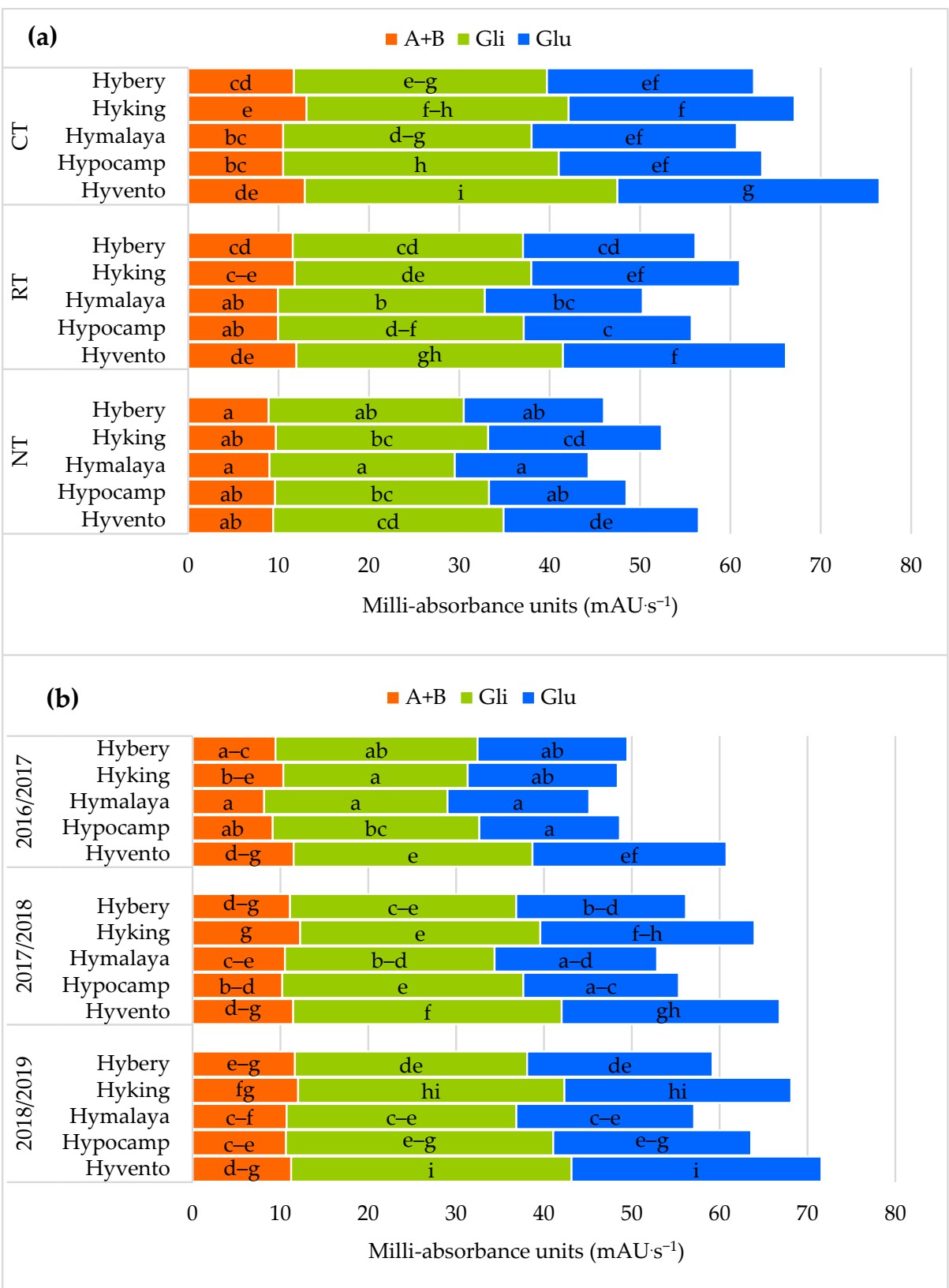

**Figure 4.** Effect of tillage systems and hybrid winter wheat cvs. on composition of protein fractions (**a**) between variants of the experiment, (**b**) between cultivars and years of research. Different letters indicate significant differences according to ANOVA (followed by Tukey's HSD test, *p* = 0.05). A + B: Albumins + Globulins; Gli: Gliadins; Glu: Glutenins.

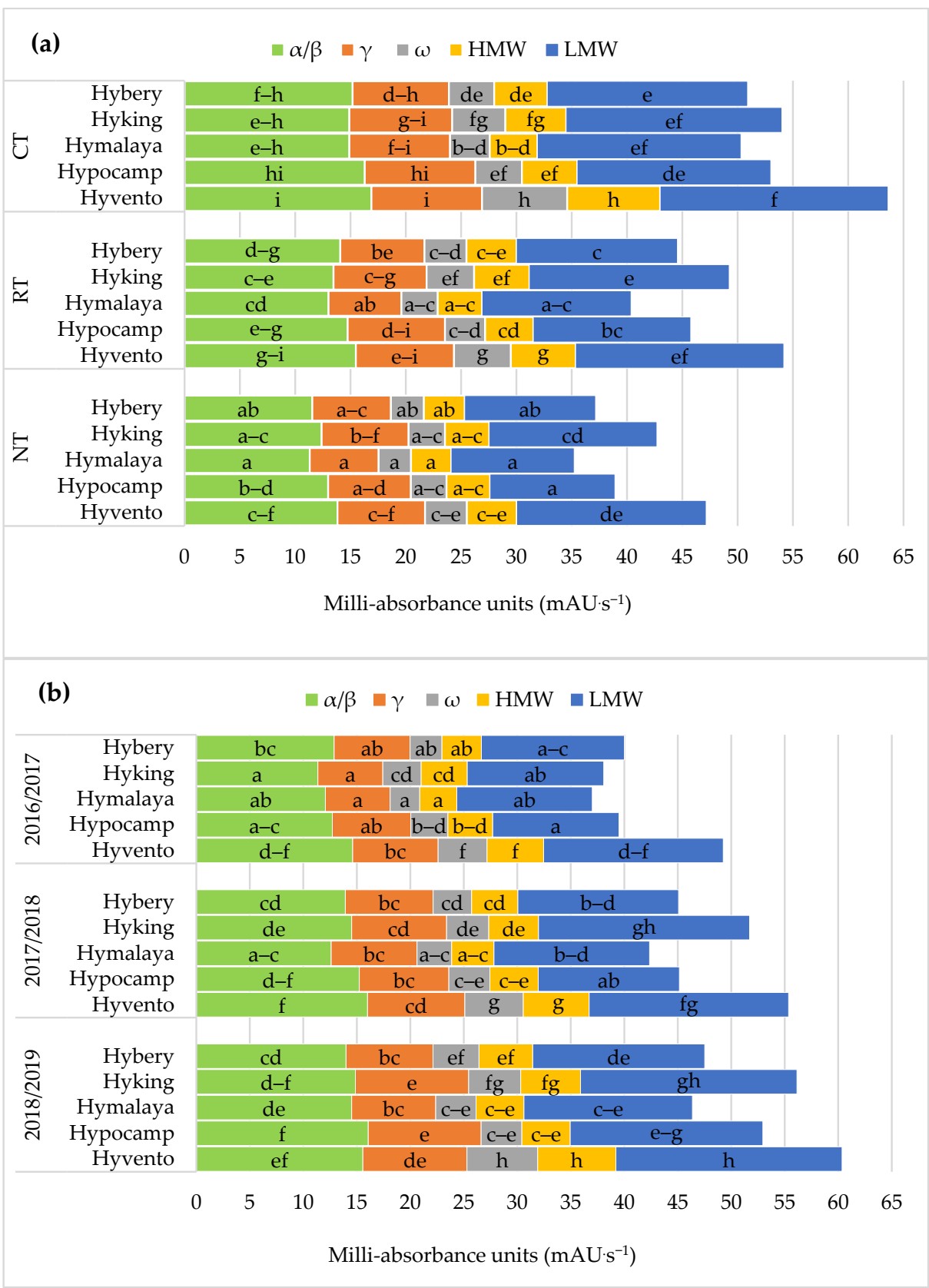

**Figure 5.** Effect of tillage systems and hybrid winter wheat cvs. on composition of subunits of protein fractions (**a**) between variants of the experiment, (**b**) between cultivars and years of research. Different letters indicate significant differences according to ANOVA (followed by Tukey's HSD test, *p* = 0.05).; α/β: α/β-gliadins; γ: γ-gliadins; ω: ω-gliadins; HMW: high molecular weight; LMW: low molecular weight.

## 4. Conclusions

The genotypic diversity of cultivars and variability of hydrothermal conditions in the years of research and their interactions with tillage systems had a decisive influence on the values of plant physiological parameters, grain yield and quality traits of hybrid winter wheat. More favourable hydrothermal conditions in the research years (2017/2018) resulted in higher grain yield, LAI values, chlorophyll content and fluorescence, and gas exchange parameters in the CT system. In the case of rainfall deficiencies (2018/2019), higher grain yield and water use efficiency ratio (WUE) were found in the RT and NT systems, with the highest values of grain and flour quality parameters and protein fractional composition in the CT system. A significant increase in the quality characteristics of grain and total gliadin and glutenin subunits and $\gamma$, $\omega$ gliadin and HMW glutenin were found in the CT system compared to RT and NT. In the CT and RT systems, no statistical differences were found in the content of albumin and globulin as well as $\alpha/\beta$ subunits of gliadins and LMW glutenins. Due to better values of physiological parameters and higher grain yield, cvs. Hymalaya and Hypocamp can be recommended for cultivation in the CT system and cv. Hyking in the NT system. Moreover, in the CT, RT and NT systems, cvs. Hyking and Hyvento obtained a grain quality higher than the other cultivars, with a more favourable fractional composition of gluten proteins. The conducted research has shown that hybrid winter wheat cultivars can be an alternative to population wheat cultivars, especially in changing agrotechnical conditions and in regions exposed to various hydrothermal conditions during the growing season.

**Author Contributions:** Conceptualization, J.B. and M.J.-P.; methodology, J.B. and M.J.-P.; investigation, M.J.-P. and J.B.; writing—original draft preparation, J.B. and M.J.-P.; writing—review and editing, J.B., M.J.-P. and D.M. All authors have read and agreed to the published version of the manuscript.

**Funding:** This work was financed by the program of the Minister of Science and Higher Education named "Regional Initiative of Excellence" in the years 2019–2022, project number 026/RID/2018/19, the amount of financing PLN 9 542 500.00.

**Institutional Review Board Statement:** Not applicable.

**Informed Consent Statement:** Not Applicable.

**Data Availability Statement:** The data presented in this study are available on request from the corresponding author.

**Conflicts of Interest:** The authors declare no conflict of interest.

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
