# Peer review of "Effect of Soil Tillage Practice on Photosynthesis, Grain Yield and Quality of Hybrid Winter Wheat"

_agriculture, doi:10.3390/agriculture11060479_

Round 1

Reviewer 1 Report

Manuscript titled "Effect of Soil Tillage Practice on Photosynthesis, Grain Yield and Quality of Hybrid Wheat" provides valuable information regarding the effect of different tillage practices, which are an essential and inseparable part of wheat cultivation. The manuscript is overall well written, and study design is solid, and the results are encouraging. 

Author Response

Response to Reviewer 1 Comments

The authors wish to thank the Reviewer for the completed review.

Reviewer 2 Report

The manuscript entitled "Effect of Soil Tillage Practice on Photosynthesis, Grain Yield and Quality of Hybrid Wheat" is a very comprehensive study, written at a good quality level.

At present, in times of climate change, but especially extreme drought during the growing season, it is necessary to look for ways to prevent evaporation from the soil while maintaining the productive and physiological functions of crops.

I have several formal comments on the manuscript, which I have marked in the text (see the attachment).

In addition, the authors missed the evaluation of the interrelationships between physiological and production parameters. I think it is interesting to note that while the "Tillage" factor showed a positive correlation between the parameters, the "Cultivar" factor showed a negative dependence between physiology, quantity compared to qualitative indicators.

Best regards,

reviewer.

Author Response

Response to Reviewer 2 Comments

Thank you for the constructive and informative comments of the Reviewer. Below are our point-by-point responses to the concerns raised during review process.

Point 1: L: 18-19 I recommended adding concrete season, because not every year were unfavourable conditions

Response 1: According to Reviewer’s suggestion we added information about growing seasons in L: 19.

Point 2: Remove dot between the units. Check whole manuscript.

Response 2: We removed dots in whole text of manuscript according to Reviewer’s suggestion.

Point 3: L:117 “Autumn would be better”

Response 3: According to Reviewer’ s suggestion we put autumn instead of fall. L: 119

Point 4: L: 279 The sentence must be reworded. The highest value can only be for one year. For two years ony in the case of the same values, which did not happen. 

Response 4:  We reworded the sentence “In the case of the Fv/Fm and PI parameters, the values were the highest in both the 2017/2018 and 2016/2017 seasons” to „In the case of the Fv/Fm and PI parameters, the values were high in both the 2017/2018 and 2016/2017 seasons” L: 303

Point 5:.L: 316 There were some other indicators that reached higher values of PN in figure 3 f. Please check it.

Response 5: According to Reviewer’s suggestion we changed “The highest values of PN, gs and E were obtained by cvs. Hymalaya, Hypocamp and Hyvento in the CT system” to “The highest values of PN, gs and E were obtained by cvs. Hymalaya, Hypocamp and Hyvento in the CT system and PN by cvs. Hymalaya and Hypocamp in the CT system”. L: 339-340

Point 6: L: 341, 349, 357, 363, - incorrect figures numeration

Response 6: We changed figures numeration according to Reviewer’s suggestions. L: 365, 373, 381, 388.

Point 7: L: 449 I suggest marked this chapter as 3.3.1.

Response 7: According to Reviewer’s suggestion we marked chapter “Proteins Composition and Characterisation” as 3.3.1. L: 478

Point 8: L: 594 – incorrect chapter numeration

Response 8: According to Reviewer’s suggestion we changed numeration of the “Conclusions” chapter. L: 615

Point 9:. In addition, the authors missed the evaluation of the interrelationships between physiological and production parameters. I think it is interesting to note that while the "Tillage" factor showed a positive correlation between the parameters, the "Cultivar" factor showed a negative dependence between physiology, quantity compared to qualitative indicators.

Response 9: In the analysis of the Results and Conclusions, it was mentioned which of the cultivars indicate better physiological and which qualitative parameters. However, the authors did not carry out a statistical analysis of the correlation between these parameters. This valuable remark will be taken into account in future publications in this field.

Reviewer 3 Report

Dear Authors,

your study is well prepared and the analytical methods are correctly used.

The study has a character of unique feature as it combines the yield data  with physiological parameter - measured in the field experiment.

Some comments I want to give for improvement:

Change "wheat" to "winter wheat". Also the title should be considered like:

“Effect of soil tillage systems on photosynthesis, grain yield  and grain quality of hybrid winter wheat.”

It is not clear form the experimental design, if winter wheat is grown in a crop rotation. It is mentioned that winter soil seed-rape was the previous crop. Was it each year?

It is important to mention also the date or date range, when the physiological parameters were measured in the field.

2.5.1. Mention also the analytical method for grain N concentration determination.

Table 4: There is a contradiction in the significance of cultivar effect on PI. ANOVA shows n.s. but the subscript of the cultivars shows different letters.

For consistency of result presentations, I recommend also to give the grain yield results in figure 4 into a table (like 4 for the physiological parameters or table 5 for quality parameter). I suppose there are also tillage x year, tillage x cultivar and cultivar x year interactions, which should be shown separately.

Table 6 should come before Figure  4 and 5.

Table 6: The composition of the protein fractions should be described in the footnote.

Author Response

Response to Reviewer 3 Comments

Thank you for the constructive and informative comments of the Reviewer. Below are our point-by-point responses to the concerns raised during review process.

Point 1: Change "wheat" to "winter wheat". Also the title should be considered like: “Effect of soil tillage systems on photosynthesis, grain yield  and grain quality of hybrid winter wheat.”

Response 1: According to Reviewer’s comments we changed “wheat” to “winter wheat” in title and whole text of manuscript.

Point 2:. It is not clear form the experimental design, if winter wheat is grown in a crop rotation. It is mentioned that winter soil seed-rape was the previous crop. Was it each year?

Response 2: In our experiment the winter oilseed rape was previous crop in all years of experiment. According to Reviewer’s suggestion we added information about pervious crop. L: 111

Point 3: It is important to mention also the date or date range, when the physiological parameters were measured in the field.

Response 3: In line 175 we added information about date range of physiological parameters measurement in the field.

Point 4: 2.5.1. Mention also the analytical method for grain N concentration determination.

Response 4: The grain N determination were performed by Kjeldahl method.

Point 5: Table 4: There is a contradiction in the significance of cultivar effect on PI. ANOVA shows n.s. but the subscript of the cultivars shows different letters.

Response 5: We have inserted the wrong significance level by mistake. The correct entry is in the Table 4.

Point 6: For consistency of result presentations, I recommend also to give the grain yield results in figure 4 into a table (like 4 for the physiological parameters or table 5 for quality parameter). I suppose there are also tillage x year, tillage x cultivar and cultivar x year interactions, which should be shown separately.

Response 6: According to Reviewer’s suggestion we gave grain yield results in Table 4 instead Figure 4a.

Point 7: Table 6 should come before Figure  4 and 5.

Response 7: According to Reviewer’s suggestion we put Table 6 before Figures 4 and 5.

Point 8:. Table 6: The composition of the protein fractions should be described in the footnote.

Response 8: According to Reviewer’s suggestion we described the composition of the protein fraction in the footnote (Table 6, L: 544).
